Computer vision supports primary access to meat by early Homo 1.84 million years ago

Cobo-Sánchez Lucía 1 2
Pizarro-Monzo Marcos 1
Cifuentes-Alcobendas Gabriel 1 3
Jiménez García Blanca 1 3
Abellán Beltrán Natalia 1 4
Courtenay Lloyd A. 5
Mabulla Audax 6
Baquedano Enrique 1 7
Domínguez-Rodrigo Manuel m.dominguez.rodrigo@gmail.com 1 3 8
1 Institute of Evolution in Africa, University of Alcala , Madrid , Madrid , Spain
2 Institute of Archaeology, University of Cologne , Cologne , Germany
3 Area of Prehistory (Department History and Philosophy, University of Alcala , Alcala de Henares , Madrid , Spain
4 Department of Artificial Intelligence of UNED (National University for Distance Education), UNED , Madrid , Spain
5 Department of Cartographic and Terrain Engineering, Superior Polytechnic School of Ávila,, University of Salamanca , Avila , Spain
6 Department of Archaeology and Heritage Studies, University of Dar es Salaam , Dar es Salaam , Tanzania
7 Regional Paleontological and Archaeological Museum of Madrid , Alcala de Henares , Madrid , Spain
8 Department of Anthropology, Rice University , Houston , TX , United States of America
Badenhorst Shaw
Electronic publication date: 2022 Oct 18
Publication date: 2022
Volume: 10
Electronic Location ID: e14148
Received 2022 Jun 2; Accepted 2022 Sep 7
Copyright: ©2022 Cobo-Sánchez et al.
Copyright year: 2022
Copyright holder: Cobo-Sánchez et al.
License: This is an open access article distributed under the terms of the Creative Commons Attribution License, which permits unrestricted use, distribution, reproduction and adaptation in any medium and for any purpose provided that it is properly attributed. For attribution, the original author(s), title, publication source (PeerJ) and either DOI or URL of the article must be cited.
License URL: https://creativecommons.org/licenses/by/4.0/

Keywords: Computer vision, Taphonomy, Tooth marks, Paleoanthropology, Meat-eating, Human evolution

Funding: The Spanish Ministry of Science and Innovation PID2020-115452GB-C21 Funding was provided by the Spanish Ministry of Science and Innovation (grant: PID2020-115452GB-C21). The funders had no role in study design, data collection and analysis, decision to publish, or preparation of the manuscript.

==============================
Human carnivory is atypical among primates. Unlike chimpanzees and bonobos, who are known to hunt smaller monkeys and eat them immediately, human foragers often cooperate to kill large animals and transport them to a safe location to be shared. While it is known that meat became an important part of the hominin diet around 2.6–2 Mya, whether intense cooperation and food sharing developed in conjunction with the regular intake of meat remains unresolved. A widespread assumption is that early hominins acquired animal protein through klepto-parasitism at felid kills. This should be testable by detecting felid-specific bone modifications and tooth marks on carcasses consumed by hominins. Here, deep learning (DL) computer vision was used to identify agency through the analysis of tooth pits and scores on bones recovered from the Early Pleistocene site of DS (Bed I, Olduvai Gorge). We present the first objective evidence of primary access to meat by hominins 1.8 Mya by showing that the most common securely detectable bone-modifying fissipeds at the site were hyenas. The absence of felid modifications in most of the carcasses analyzed indicates that hominins were the primary consumers of most animals accumulated at the site, with hyenas intervening at the post-depositional stage. This underscores the role of hominins as a prominent part of the early Pleistocene African carnivore guild. It also stresses the major (and potentially regular) role that meat played in the diet that configured the emergence of early Homo.

Introduction

For more than five decades, archaeologists have debated if the earliest members of our genus were hunters or scavengers (Potts, 1988; Blumenschine, 1991; Blumenschine, 1995; Capaldo, 1997; Selvaggio, 1998; Domínguez-Rodrigo & Barba, 2006; Domínguez-Rodrigo et al., 2005; Domínguez-Rodrigo, Barba & Egeland, 2007; Domínguez-Rodrigo et al., 2014; Domínguez-Rodrigo et al., 2015; Domínguez-Rodrigo et al., 2021a; Domínguez-Rodrigo et al., 2021b; Dominguez-Rodrigo & Pickering, 2017; Gidna et al., 2014; Organista et al., 2015; Organista et al., 2016; Parkinson, 2018). Hunting would have enabled continuous access to high-quality animal resources, whereas scavenging would have been mostly feasible seasonally and on specific ecotones, yielding most frequently flesh scraps and long bone marrow (Blumenschine, 1986; Gidna et al., 2014). The evolutionary relevance of both strategies cannot be overemphasized. Hunting could have triggered complex hominin behaviors, including high degrees of cooperation and intentional food-sharing, which are at the root of what defined humans as opposed to other primates (Isaac, 1978). Passive scavenging would not necessarily stimulate these behaviors (Potts, 1988; Blumenschine, 1991). Hunting would also have provided a regular source of high-quality food, which would have prompted the transfer of basal metabolic energy from the digestive tract to the brain enabling its evolution (Aiello & Wheeler, 1995; but see Navarrete, Van Schaik & Isler, 2011). Additionally, the regular pursuit of mobile foods would have increased home range sizes and impacted hominin spatial behavior (Wood et al., 2021). Ecologically, it has also been argued that the genus Homo started to impact African biomes by 2 Ma by outcompeting other carnivores (Werdelin & Lewis, 2013; Faurby et al., 2020). If hominins entered the predatory guild at that time, this may have cascaded through the food chain resulting in niche overlap and eventual wiping of other carnivores, as reflected in the paleontological record for this period (Werdelin & Lewis, 2013). This might have been the first recorded impact on humans at an ecosystemic scale; however, objective empirical proof thereof was missing until now.

A broad range of questions centered on early hominin socio-ecology, thus, depends on whether archaeologists can assess how hominins acquired and consumed carcasses. Fortunately, this information is contained in exceptionally well-preserved faunal assemblages from the earliest sites of Olduvai Gorge (Tanzania), and advanced technological innovation (like the artificially intelligent methods applied here) provide an excellent opportunity to understand hominin socio-economic behavior in its ecological context.

The earliest uncontroversial evidence of hominin meat-eating dates to 2.6 Ma (De Heinzelin et al., 1999; Domínguez-Rodrigo et al., 2005), together with the earliest uncontroversially-dated stone tools (Semaw et al., 1997). Claims about Pliocene archeological traces (McPherron et al., 2010; Harmand et al., 2015) do not currently withstand close scrutiny (Domínguez-Rodrigo & Pickering, 2010; Domínguez-Rodrigo & Alcalá, 2016; Dominguez-Rodrigo & Alcalá, 2019; Domínguez-Rodrigo, Pickering & Bunn, 2010). Although the heuristics of the arguments for interpretations of hunting and scavenging are asymmetrical (Domínguez-Rodrigo, Barba & Egeland, 2007; Domínguez-Rodrigo, 2009), there is a widespread consensus that the question is unresolved. Advocates of passive scavenging models have posited that hominins were preceded by felids in carcass consumption, then followed by hyenids (Blumenschine, 1995; Capaldo, 1997; Selvaggio, 1998). The only evidence provided for this would be the purportedly high tooth mark frequencies documented on long bone shafts from the 1.84 Ma site of FLK Zinj (Olduvai Gorge) (Blumenschine, 1995; Capaldo, 1997; Selvaggio, 1998). Nevertheless, these frequencies span a range of tooth marks between 300%–500% higher than documented among modern felid-modified fragmented long bone assemblages (Domínguez-Rodrigo & Barba, 2006; Domínguez-Rodrigo, Egeland & Pickering, 2007; Organista et al., 2016; Domínguez-Rodrigo et al., 2021c). Subsequent re-examination of the Zinj assemblage led some authors to document that a high proportion of such marks were biochemical bioerosive modifications (Domínguez-Rodrigo & Barba, 2006; Parkinson, 2018). This controversy over the agency in carcass consumption sequence at early sites would be solved if carnivore-specific identifications were made confidently using the tooth mark evidence from these early assemblages. Here, we implement a method that successfully and objectively differentiates tooth marks made by lions and hyenas. This enables the analyst to determine more accurately the input of both agents in early sites, and the role of hominins in the carcass acquisition and consumption process. If hominins were preceded by lions in carcass consumption, we would expect the tooth mark signals on meat-bearing long bone shafts to be diagnostically felid. If, instead, hominins had primary access and hyenas intervened postdepositionally, then a hyenid signal on the same bone portions would be expected.

Recent methodological developments in taphonomy based on the use of tools in artificial intelligence (AI) and computer vision (CV) have been made, presenting objectively high resolution in the differentiation between bone-modifying agents. These advances have even been able to accurately (>95%) discern cut marks produced on bones containing bulk flesh as well as those produced when bones are mostly defleshed (Cifuentes-Alcobendas & Domínguez-Rodrigo, 2019; Domínguez-Rodrigo et al., 2020). Convolutional Neural Network architectures fine-tuned using transfer learning techniques have also been successfully applied to carnivore tooth marks, reaching up to 89% testing accuracy for the classification of lions and jaguars; two carnivores with similar morphological dentition and extremely similar tooth marking features (Jiménez-García et al., 2020a; Jiménez-García et al., 2020b). The same methods have also enabled taphonomists to differentiate >92% of tooth scores made by lions and hyenas (Abellán et al., 2021). Here, we use the same method, which classified successfully 100% of the testing sets of experimental tooth pits made by lions and hyenas, by additionally applying it to the tooth mark samples of the 1.84 Ma DS site, Bed I (Olduvai Gorge). This is the first time that these methods have been applied to a prehistoric assemblage.

DS (David’s Site) is the largest documented early Pleistocene Oldowan site (560 m2) and it lies on the same thin clay stratum as FLK Zinj (Domínguez-Rodrigo et al., 2017; Diez-Martín et al., 2021) (Fig. S1). It is also capped by the same 1.84 Ma Tuff IC (Arráiz et al., 2017). Recent taphonomic analyses of the site’s faunal assemblage recovered from Level 22B showed a distribution of cut marks on long bones that only matched experimental assemblages reproducing primary access to fully fleshed carcasses (see Supplementary Information, Figs. S5 and S6) (Domínguez-Rodrigo et al., 2021a). Meat-bearing bone specimens were cut-marked at an average of 10.2%. Long bones were also systematically demarrowed by hominins as evidenced by the frequency (6.7%) and distribution of percussion marks (see Fig. S4). Tooth mark frequencies are very low (1.9%) and also correspond to experimental scenarios of carnivores having access to hominin hammerstone-broken bones. Nevertheless, and contrary to what was initially argued (Selvaggio, 1994; Blumenschine, 1995), lions leave very few tooth marks on long bone shafts from the carcasses that they consume, especially if bones are subsequently fragmented by humans to extract marrow (Gidna et al., 2014; Organista et al., 2016). For this reason, low frequencies of tooth marks on long bone shafts could be indicative of felid primary access or hyena secondary access to bones. Differentiating between both agents is crucial. Here, we will study the 35 tooth marks (15 tooth scores and 20 tooth pits) documented on the bone specimens from DS and provide identification of the carnivore agent(s) involved and their impact.

Methods

The hypothesis to be tested here is the following: if the hominins responsible for the formation of DS obtained carcasses from felid kills, the resulting accumulation of bones will bear traces of lions on element portions that remain defleshed after felid consumption (Gidna et al., 2014; Parkinson, Plummer & Hartstone-Rose, 2015; Domínguez-Rodrigo et al., 2021c). These are namely long bone mid-shafts, which are the best preserved portions in most anthropogenic archaeofaunal assemblages and at DS (Marean, 1991; Arriaza et al., 2019; Arilla, Rosell & Blasco, 2019). In a felid-hominin-hyenid model, midshafts will be toothmarked by felids if they precede hominins. Hominin hammerstone breakage of long bones would result in limited impact by ravaging hyenas given the lack of nutrients on these discarded portions, but these would be the only agent leaving tooth marks there (other than the unexplored option of hominin themselves), since felids would be uninterested in modifying these fragmented bone portions given their lack of flesh.

Sample

The lion-hyena tooth sample used here was obtained from experiments carried out with captive carnivores at the reserve of Cabárceno (Cantabria, Spain) and in the wild at the Tarangire National Park (Tanzania). At Cabárceno, carnivores live in open spaces and they do not undergo the stereotypic behaviors that carnivores display in cages or in small enclosures (Gidna, Domínguez-Rodrigo & Pickering, 2015). Carnivores in Cabarceno live in large areas comprising several thousands of square meters (http://www.parquedecabarceno.com). Consumption animals were provided from a butchery company complying with all sanitary protocols and with whom the Cabárceno reserve has a MOU. Carcass parts consumed by lions were collected after a few days of exposure (when they were completely defleshed and unattended, which usually spanned 1–4 days). With hyenas, the protocol was modified, because when bones were exposed for more than one day, they tended to be completely consumed. Thus, bones in their enclosure were collected earlier, usually on the same day, after a few hours of consumption. After that, bones were cleaned with a solution of neutral detergent, then left to dry.

The lion tooth mark sample was obtained from the experiment reported by Gidna, Yravedra & Domínguez-Rodrigo (2013), consisting of 112 limb bones from 18 zebra carcasses altered by wild lions from the Tarangire National Park plus 60 limb bones from nine older juvenile and prime adult horses consumed by 11 semi-captive lions from Cabárceno private reserve. Animals from Tarangire were obtained from lion kills under the research permit provided by Tanzanian National Parks (TANAPA). Consumption animals from Cabárceno were obtained from the butchery provider described above.

The spotted hyena bone collection was also obtained at the Cabárceno reserve in Cantabria (Spain). The sample analyzed was composed of 67 long bones from 23 carcasses from adult horses consumed by a variable number of hyenas.

The modern lion and hyaena tooth mark image dataset used to train our models includes a total of 355 tooth marks. Of these, 195 are scores made by lions and 80 scores made by hyaena. In addition, there are 35 pit marks caused by lions and 45 pits by hyaena. We subsequently test our models on the images of 35 tooth marks (15 scores and 20 pits) recorded on bones from DS. Two different sets of models were made for the tooth scores and the tooth pits.

The DS fossil sample was collected from excavations at Olduvai Gorge (Tanzania) by the Olduvai Paleoanthropology and Paleoecology Project (TOPPP) under the COSTECH and Antiquities permit number 2021-631-NA-2006-115.

Deep learning analysis

Here, some of the most advanced classification algorithms that exist for classifying images were used. Deep Learning (DL) methods that have successfully competed in image classification were implemented. Complex convolutional neural networks (CNN) from transfer learning (i.e., pre-trained architectures on thousands of diverse objects) were applied. Given the success of previous models in studies using CNN for bone surface modification (BSM) classification (Cifuentes-Alcobendas & Domínguez-Rodrigo, 2019; Domínguez-Rodrigo et al., 2020; Jiménez-García et al., 2020a; Jiménez-García et al., 2020b; Abellán et al., 2021), in the present study two different model architectures successful in those experiments were used: ResNet 50 (version 1.0) and VGG19 (He et al., 2016; Simonyan & Zisserman, 2014a; Simonyan & Zisserman, 2014b). Given the original small sample sizes, the architectures were used with image augmentation. For this purpose, samples were augmented via random transformations of the original images involving shifts in width and height (20%), in shear and zoom range (20%), and also including horizontal flipping, as well as a rotation range of 40°. Some of these models had previously been applied to a set of tooth scores from five different carnivore types, which included lions, jaguars, spotted hyenas, crocodiles and wolves (Abellán et al., 2021). This comparative study was published separately. Here, we used only lions and hyenas because the other carnivore mammals do not apply, since they are absent in the African biomes, and the conspicuous damage documented on carcasses at these early sites do not exhibit any crocodile damage. Not a single bisected tooth mark was documented at DS and PTK, and these distinctive types of tooth marks occur in >80% of bones modified by these reptiles (Baquedano, Domínguez-Rodrigo & Musiba, 2012). Lions were selected also because they were the most likely predator enabling secondary access to their kills if hominins were scavenging medium-sized carcasses, which are predominant at both Olduvai sites. Hyenas were selected because they are the most likely agents modifying bones if the role of non-hominin carnivores is restricted to secondary access to carcasses after having been consumed by hominins. Since these are the two hypotheses that we intended to test, the selected carnivores are the adequate ones to carry out the testing.

We are aware that modern lions and hyenas do not necessarily represent the exact tooth morphology of Early Pleistocene lions and hyenas. An on-going study using 3D artificially-created tooth marks using prehistoric and extinct mammal carnivores will elucidate the similarities and differences among extinct and extant carnivore taxa (Domínguez-Rodrigo et al., in progress). In the meantime, we assume that little variation exists between tooth mark morphologies of modern and prehistoric hyenas, as well as modern and prehistoric lions. If such variation existed, we assume that extinct hyenas would have had tooth morphologies (which ultimately determine tooth mark morphologies) more similar to modern hyenas than to modern lions. In the Early Pleistocene, Crocuta crocuta is not present, but two ancestral forms are: Crocuta ultra and the earlier lineage Crocuta dietrichi (Werdelin & Lewis, 2005). A proof that the first assumption is well-supported lies on the metrics of the premolars (the most impacting teeth during bone breaking and marking) of the Crocuta ultra hypodigm (similarly sized to modern spotted hyenas) and an extended sample of modern Crocuta crocuta (Lewis & Werdelin, 2022). Additionally, most of the Olduvai Bed I hyena dental remains show metrics, especially p3 and p4, that fall well within the range of variation of modern Crocuta crocuta (Lewis & Werdelin, 2022). Tooth marks made by lions would not represent such a problem, since Panthera leo has also been identified in the Early Pleistocene, most specifically at Olduvai during Bed I (Werdelin & Lewis, 2005). Lion tooth marks, thus, would be very similar if not the same during Bed I.

The lion-hyena tooth mark classifications were carried out using the two models (ResNet 50 and VGG19), fine-tuned using transfer learning approaches. In each of the models used, the activation function for every layer was a rectified linear unit (ReLU). The last fully connected layer of the network used a “sigmoid” activation for the binary comparison between lions and spotted hyenas. The loss function selected was binary cross-entropy. Cross-entropy measures distances between probability distributions and predictions (Chollet, 2017). The optimizer used was Stochastic Gradient Descend (SGD) with a learning rate of 0.001 and a momentum of 0.9. Accuracy was the metric selected for the classification process.

The models were trained on 75% of the original image dataset. The resulting models were subsequently tested against the 25% remaining sample, which was not used during the training. Training was performed through mini-batch kernels (size = 32–20 for tooth scores and tooth pits respectively). Testing was made using mini-batch kernels of size 20-10 (for tooth scores and tooth pits respectively). Weight update was made using a backpropagation process for 100 epochs. Images of BSM were produced with a binocular microscope (Optika) at 30 X using the same light intensity and angle. The resulting image data bank was used for analysis through the CNN models described above. All images were transformed into black and white during image processing in the Keras Application Programming Interface (API), by using bidimensional matrices for standardization and centering. An important difference from previous analyses on the same dataset was that standardization was carried out using each model’s preprocessing centering functions. Batch sizes as well as steps per epoch also differed (Abellán et al., 2021). Each image was then reshaped so that they share the same dimensions (80 × 400 pixels for tooth scores and 150 × 200 for tooth pits). The Keras (2.4.3) library was used with the TensorFlow (2.3.0) backend. Computation was carried out with A HP Z6 workstation, using a CUDA computing (cuDNN) environment.

The classification of the DS tooth marks was carried out using the two models applied to the experimental data set (ResNet50 and VGG19). Marks were interpreted as agent specific only when the two models coincided in the determination and, at least one of them, yielded classification probabilities >70%. Marks in which one of the models yielded a different agent classification were discarded for the interpretation, because they would not be reliable. A majority voting approach was adopted by comparing the results of both models. Mark probabilities were derived from the sigmoid function, which provides the probability that the input data represent the positive class, which in this case was “lion tooth mark”. Probabilities for the positive class were the unmodified estimates of the sigmoid function. Probabilities for the negative class (here, “hyena tooth mark”), were derived from the reverse of the sigmoid output. This protocol is common in application of machine learning algorithms to binary problems (Kuhn & Johnson, 2013). The experimental image data base and code can be found in the public repository: https://dataverse.harvard.edu/dataset.xhtml?persistentId=doi%3A10.7910%2FDVN%2FBQTKBA.

Training graphs displaying accuracy and loss during training and validation were also used to assess over-underfitting processes. The models were run twice. One without Dropout and the other with Dropout (0.3). This was intended to detect also over- and under-fitting behaviors of the transfer model architecture as fit to data. Dropout did not introduce any significant difference with the models (See Supplementary Information). The training process was also graphically documented using both methods (Fig. 1). As one of the models (VGG19) applied to the tooth scores yielded marginal overfitting both with and without use of the Dropout generalization method, we decided to run all models for tooth scores also including Early Stopping (Goodfellow, Bengio & Courville, 2016; Brownlee, 2017; Brownlee, 2019). In this case, Early Stopping was combined with Dropout. Given that Dropout has the potential of interfering with Early Stopping, when the percentage of nodes discarded by Dropout is high, we lowered Dropout to 0.2. The metric used for Early Stopping was accuracy. The mode used was maximization of accuracy. In order to avoid local minima, we added a patience value of 15 epochs. The best model was automatically saved. The models did not improve using this combination of regularization techniques. This is why we did not use the models combining Dropout and Early Stopping for the analysis of the DS tooth marks. We show the results in the Supplementary Information.

Figure 1 Accuracy (upper) and loss (lower) of the Resnet50 model on the tooth pit subsample.

Graphs show the transfer learning model without (left) and with Dropout (right). See Supplementary File for data on precision, recall, F1-scores,micro-and macro- average accuracy.

It has been argued that saliency methods are adequate to detect what DL models focus on when classifying images (Simonyan, Vedaldi & Zisserman, 2014). Saliency is a concept that refers to specific features that depict identifying locations within an image. A saliency map is a topographical representation of such features. Saliency maps can be generated from every convolutional layer within a DL network, but they usually are made based on the last convolutional layer prior to the flattening process. There are several types of saliency map algorithms, but in our models, a gradient visualization for detecting the features that influenced discrimination was applied using a gradient weighted activation mapping algorithm (Grad-CAM) (Selvaraju et al., 2016; Selvaraju et al., 2017). This method overlays a heatmap on the original image based on gradients of the predicted class derived from the last convolutional feature map. The Grad-CAM algorithm highlights areas of the marks that are most important for the prediction and classification of the image (Cifuentes-Alcobendas & Domínguez-Rodrigo, 2019). The use of this algorithm showed the importance of certain areas for correct image classification. Here, we applied it to a selection of tooth marks, in order to compare if there was convergence in the saliency maps produced by Resnet50 and VGG19.

Results

Both the Resnet50 and VGG19 models displayed a regular training process for the tooth pit samples, with both classes clearly differentiated in the early stage of the process. Overfitting (or high variance) was not an issue (despite the early obtainment of high accuracy in the training process), since the accuracy of the training set is not substantially different from that of the validation set, and the shape of both during the training process is similar (Figs. 1 and 2). This is also supported by the loss values. Overfitting should be inferred if loss values for the training set were low, but high in the validation/testing, as well as displaying both sets differences in their shape and longitudinal values. This is not documented in the samples used, where in all cases the training loss is undifferentiated from the validation loss (Figs. 1 and 2). This is also documented for the tooth score sample, where both models showed similar trajectories for the training and validation subsamples (Figs. 3 and 4). Only in the case of the VGG19 model applied to tooth scores can a marginal overfitting effect be observed in the tooth score sample (Fig. 4), which may only have impacted its higher confidence regarding the Resnet50 model in the probability values of classification, since both models yielded the same classification for virtually almost all of the tooth scores. Running the models on the tooth scores with the combination of Dropout and Early Stopping did not improve the accuracy (Supplementary Information). All of them produced lower accuracy values than the use of the transfer learning model without any of these regularization methods. The Early Stopping-Dropout (ESDO) combination yielded also models with poorer training than when using Dropout alone.

Figure 2 Accuracy (upper) and loss (lower) of the VGG19 model on the tooth pit subsample.

Graphs show the transfer learning model without (left) and with Dropout (right). See Supplementary File for data on precision, recall, F1-scores,micro-and macro- average accuracy.

Figure 3 Accuracy (upper) and loss (lower) of the Resnet50 model on the tooth score subsample.

Graphs show the transfer learning model without (left) and with Dropout (right). See Supplementary File for data on precision, recall, F1-scores,micro-and macro- average accuracy.

Figure 4 Accuracy (upper) and loss (lower) of the VGG19 model on the tooth score subsample.

Graphs show the transfer learning model without (left) and with Dropout (right). See Supplementary File for data on precision, recall, F1-scores,micro-and macro- average accuracy.

For the experimental data, the ResNet 50 model yielded an accuracy of 95.6% in the classification of tooth scores (94.20% if using the Dropout regularization method; 84% if using ESDO) compared to 88.41% of accuracy (85.51% if using Dropout; 67% if using ESDO), resulting from the VGG19 model (Table 1; Supplementary Information). For tooth pits, the VGG19 architecture yielded the best classification for the testing set (accuracy=100%; 95.6% if using Dropout). The Resnet 50 model yielded similarly high accuracy (95.6%; with and without Dropout) (Table 2; Supplementary Information). Both models showed a fairly balanced classification with high F1-score and AUC values (See Supplementary Information).

The application of both models to the DS tooth pit and score data, yielded unambiguous classifications of almost all of the tooth marks; only two tooth pits were discarded because both models yielded contradictory classifications. The remainder of tooth marks were classified similarly by both models, with probabilities >80% (Table 3). Table 3 shows the probability distribution for each mark. Classification was deemed highly reliable only when the probability of classification was >75%, moderately reliable (probability = 60%–75%) and unreliable (<60%).

Table 1 Accuracy, loss and F1-score and Area Under the Curve (AUC) for the CNN models (the two most successful transfer learning methods) applied to the lion-hyena tooth score marks testing set.

Model	Accuracy	Loss	F1 macro avg	AUC	
ResNet50	95.56	0.116	0.95	0.939	
VGG19	88.41	0.294	0.87	0.903	

Table 2 Accuracy, loss, F1-score and Area Under the Curve (AUC) for the CNN models applied to the lion-hyena tooth pit marks testing set.

Model	Accuracy	Loss	F1 macro avg	AUC	
ResNet50	95.65	0.096	95	0.944	
VGG19	100	0.0491	1.00	1.0	

Table 3 Anatomical element and animal carcass size of the tooth marked specimens from DS 22B and probability of classification of each tooth mark yielded by the most successful CNN models (ResNet 50 model and VGG19 transfer learning architectures).

Pits			VGG19	RESNET 50		
Label	Element	Animal size	Prob. Hyena	Prob. Lion	Prob. Hyena	Prob. Lion	Classificaton	
410	metacarpal	3	0.999	0.001	0.999	0.001	Hyena	
1020a	tibia	3	0.999	0.001	0.983	0.017	Hyena	
1020b	tibia	3	0.999	0.001	0.999	0.001	Hyena	
1422	caudal vertebra		0.999	0.001	0.384	0.616	Discarded	
1445a	metapodial	4–5	0.765	0.235	0.98	0.02	Hyena	
2476	metacarpal	3	0.999	0.001	0.999	0.001	Hyena	
2560	tibia	3	0.999	0.001	0.999	0.001	Hyena	
2590	rib	3	0.964	0.036	0.11	0.89	Discarded	
2880	humerus	1	0.814	0.186	0.979	0.021	Hyena	
2945	humerus	3	0.992	0.008	0.997	0.003	Hyena	
3121a	metatarsal	3	0.879	0.121	0.645	0.355	Hyena	
3112b	thoracic vertebra	1	0.999	0.001	0.989	0.011	Hyena	
3112c	thoracic vertebra	1	0.999	0.001	0.999	0.001	Hyena	
3125a	tibia	3	0.999	0.001	0.999	0.001	Hyena	
3125b	tibia	3	0.999	0.001	0.999	0.001	Hyena	
3492	rib	3	0.999	0.001	0.937	0.063	Hyena	
3753b	radius	3	0.999	0.001	0.999	0.001	Hyena	
4288	mandible	3	0.01	0.99	0.01	0.99	Lion	
4487	rib	2	0.999	0.001	0.971	0.029	Hyena	
1756	indet	3	0.999	0.001	0.976	0.024	Hyena	
Scores			VGG19	RESNET 50		
Label	Element	Animal size	Prob. Hyena	Prob. Lion	Prob. Hyena	Prob. Lion	Classificaton	
99	atlas	3	0.827	0.173	0.691	0.309	Hyena	
410a	metacarpal	3	0.904	0.096	0.641	0.359	Hyena	
410b	metacarpal	3	0.904	0.096	0.653	0.347	Hyena	
410c	metacarpal	3	0.915	0.085	0.664	0.336	Hyena	
1020	tibia	3	0.885	0.115	0.702	0.298	Hyena	
1422	caudal vertebra	3	0.868	0.132	0.668	0.332	Hyena	
1808	middle phalanx	3	0.889	0.111	0.709	0.291	Hyena	
2258a	carpal	4–5	0.917	0.083	0.711	0.289	Hyena	
2258b	carpal	4–5	0.851	0.149	0.706	0.294	Hyena	
2560	tibia	3	0.848	0.152	0.672	0.328	Hyena	
3116	metacarpal	1–2	0.859	0.141	0.608	0.392	Hyena	
1915	Indet limb	5–6	0.946	0.054	0.765	0.235	Hyena	
920a	Indet limb	3	0.839	0.161	0.746	0.254	Hyena	
920b	Indet limb	3	0.899	0.101	0.751	0.249	Hyena	
920c	Indet limb	3	0.863	0.137	0.718	0.282	Hyena	

From the tooth score sample, all marks were classified as hyena-made. The probabilities in all cases are >80% for the VGG19 model and ranging between 60%–76% for the ResNet 50 model. This indicates that agency attribution based on the classification of scores is fairly reliable. Likewise, the classification of the DS tooth pits has yielded high confidence, with the VGG19 and ResNet50 models showing a probability >90% in most cases. All tooth scores and all tooth pits (except one) from DS 22B are classified with high reliability as hyena tooth marks (Fig. 5). We use these for inference of carnivore agency at the site (Fig. 6). Only one tooth mark was identified as lion-made with very high confidence (probability = 99%).

Figure 5 Examples of the photographs of tooth pits from the DS 22B bone sample used for the analysis.

Figure 6 Carnivore damage documented in the DS 22B faunal assemblage and attributed to hyena ravaging.

The application of saliency maps showed that the Resnet50 and VGG19 models were identifying similar areas of the marks to classify them. Most of the identification was carried out either inside the marks (Fig. 7) or on their borders (Fig. 8). This can be explained because the microscopic features that make marks identifiable also occur on the border and mark shoulder, which is where mark delineation takes place. For example, lions make shallower tooth scores than hyenas and this is reflected in straighter score walls, whereas hyena durophagous behavior is reflected in deeper penetration and more cortical crushing, thereby resulting is more irregular wall outlines.

Figure 7 Saliency map for experimental tooth mark.

Example of lion tooth pit (upper) and saliency map produced by the Resnet50 model (lower).

Figure 8 Saliency map for archaeological tooth score according to model.

Example of DS tooth score mark (cropped image focusing on the groove) (A), and the saliency maps produced by the Resnet50 (B) and VGG19 (C) models.

Discussion

The emergence of early Homo has mainly been linked to two phenomena: encephalization and dietary change. The reduction of the dentition in some hominin taxa for the first time in four million years of evolution could be a reflection of the increase of the food quality which in a savanna biome, was probably the outcome of the adoption of carnivory. Although traditionally, the attribution of the earliest archaeological record has been made to Homo habilis, the current picture is far more complex. On the one hand, the early Homo hypodigm has been fragmented into a minimum of two species, being widely different regarding dentition and encephalization (Spoor et al., 2007; Antón, Potts & Aiello, 2014). On the other hand, the earliest record of Homo is documented at least 200 ka prior to the oldest uncontroversial archaeological record (Villmoare et al., 2015). Lastly, current evidence goes against previous interpretations of anagenetic evolution of African H. erectus (H. ergaster) from H. habilis, since the oldest dates for the former reach 2 Ma (Villmoare et al., 2015). This is reinforced by the presence of a significantly bigger and more modern hominin with H. erectus morphology in East Africa by that date (Domínguez-Rodrigo et al., 2015a; Hammond et al., 2021). Therefore, both in South and East Africa, there are at minimum of three hominin lineages co-existing. We argue that, given the behavioral complexity from the Olduvai Bed I sites, it is more parsimonious to attribute the authorship of that record to the most complex hominin existing at that time, making early H. erectus (ergaster) the most likely candidate. This is the species that we have in mind when we attribute this archeological record to early Homo sensu lato.

The high probability of classification of all DS tooth scores and pits (except one) as caused by hyenas shows that most carnivore damage is the result of the impact of durophagous carnivores after the deposition of carcass remains. The probability of hyenid impact is systematically high in most tooth marks discovered. Other taphonomic analyses reveal that the overall low frequency of carnivore damage documented at the site is attributable to hyena ravaging (Fig. 6 and Figs. S4, S7A and S7B). The fact that only hyenas are securely identified as the main bone modifiers in addition to hominins at DS supports that their intervention took place after hominins discarded the carcasses that they consumed. This is also because no resources are available for scavenging after hyena intervention (Brain, 1981; Prendergast & Domínguez-Rodrigo, 2008; Domínguez-Rodrigo & Pickering, 2010). The possibility of hyenas having independently contributed to the assemblage was also considered, but it was discarded because no hyena taphotype produced during primary access was found, and because their modified carcasses would show bone damage patterns (i.e., gnawing, static-loading overlapping and opposing notches) that was not documented at the site (Domínguez-Rodrigo et al., 2015; Domínguez-Rodrigo & Pickering, 2010). This means that hominins were not commonly acquiring carcasses after they had been defleshed by felids. Additional evidence thereof is found in the taphotype study of long bones, which also shows that typical felid bone modification patterns (Gidna, Yravedra & Domínguez-Rodrigo, 2013; Parkinson, Plummer & Hartstone-Rose, 2015; Domínguez-Rodrigo et al., 2015c; Pobiner, Dumouchel & Parkinson, 2020) are absent from the bone assemblage (Fig. S7C). Moreover, the documented taxa diversity at the site contrasts with a highly specialized felid predatory range (Arriaza & Domínguez-Rodrigo, 2016; Fig. S3).

Only one tooth mark (DS4288) shows confident classification as a lion-made tooth pit (Table 3). This is of extreme relevance because it also shows that hominins occasionally engaged in opportunistic carcass exploitation, like modern human foragers and mammal predators do (Domínguez-Rodrigo et al., 2021b). Recently, the only partial axial skeleton found at DS yielded evidence of associated limb elements modified by felids and hominins (Domínguez-Rodrigo et al., 2021b). This constitutes empirical evidence of diverse carcass acquisition strategies by hominins. The presence of cut marks on the same associated radius-ulna where the lion tooth marks were documented indicate that the carcass was not utterly defleshed upon acquisition by hominins. Given the substantially deeper profile of such tooth marks on the cancellous tissue of the ulnar olecranon, they have not been included in the present analysis, since most experimental tooth marks are documented on long bone shafts. The only tooth mark from the DS sample used here that was classified as lion-made was interestingly found on a mandible fragment in spatial association to the scavenged radius-ulna. Both were located less than two meters away from each other. This is important, because it shows that out of the 560 m2 of the excavated site, all lion-impacted bone specimens are in close spatial association. This also attests to the efficiency of the DL method to identify agency in carnivore bone damage. Additionally, this also documents that the site underwent virtually very limited post-depositional disturbance, since the original spatial association of the scavenged hominin-processed carcass remains was not altered after discard.

The overwhelming confirmation that most tooth marks at the site were made by hyenas and the lack of felid taphotypes on the preserved long bone ends show that hominins were predominantly enjoying primary access to completely fleshed carcasses. The abundance and anatomical distribution of cut marks at DS, which coincide with experimental models of primary and early access to carcasses by hominins, further supports this interpretation (Figs. S5 and S6). Also, the documented butchery pattern on long bones matches the pattern of processing fully fleshed carcasses (Domínguez-Rodrigo et al., 2021a). In fact, hominins engaged in the complete butchering process at DS, also accessing the marrow content of bones (Fig. S4).

Meat must have therefore been crucial for the adaptation of early Homo and must have had serious repercussions for early human physiology. It has been argued that changes such as the reduction in tooth size or major skeletal modifications observed in some early Homo are a reflection of the anatomical impact caused by the dietary change towards carnivory (Ungar, 2012; Zink & Lieberman, 2016; Domínguez-Rodrigo et al., 2015b; Herries et al., 2020). As a matter of fact, it has been argued that evidence of some typically-human pathologies (like porotic hyperostosis) documented in an infantile 1.5 Ma individual might be related to deficiencies in cobalamin consumption, which could be related to a meat-dependent physiology at that time (Domínguez-Rodrigo et al., 2012). Possible pathologies related to the consumption of animal protein have also been observed in other hominin taxa (D’Anastasio et al., 2009) This first direct evidence of a hominin-hyenid interaction in the modification of an early Pleistocene assemblage also shows that the significant increase in brain size documented after 2 Ma co-occurs with this evidence of bulk flesh consumption by hominins.

The present study also suggests that by having primary access to fleshed carcasses hominins may have already entered the predatory guild at that time if not earlier. Regular hunting of small and medium-sized animals by early Homo may have had a significant ecological impact on the carnivore predatory guild. As a matter of fact, the Pliocene and Pleistocene in East Africa witnessed a significant carnivore diversity decrease (>99% loss of functional richness) (Werdelin & Lewis, 2013). Studies have shown that these carnivore extinctions do not correlate with climatic and environmental factors. In contrast, they strongly correlate with hominin brain expansion (Faurby et al., 2020). This suggests that an increase in cognitive capacities may have enabled early Homo to access and even overtake the ecological niches of other carnivores and that anthropogenic impact on biodiversity started much earlier than previously thought (Faurby et al., 2020). The new high-quality omnivorous diet likely led to hominin population expansion across landscapes, placing early Homo at selective advantage over other competing carnivores. Evidence thereof is found in the exploitation of megafauna and in the much larger Acheulian sites after 1.7 Ma (Domínguez-Rodrigo et al., 2014; Organista et al., 2015; Organista et al., 2017). Given the ecological correlation between prey size exploitation and number of carnivores involved in meat consumption (Portalier et al., 2019; Vézina, 1985; Loveridge et al., 2009; Tsai, Hsieh & Nakazawa, 2016; De Cuyper et al., 2019), access to very large ungulates by hominins likely indicates larger hominin groups.

Killing large animals is traditionally identified as one of the fundamental characteristics of human predatory behavior (when compared to the hunting behavior of other primates) and it illustrates the acquisition of large amounts of meat (Agam & Barkai, 2018). This activity commonly requires collective participation of several individuals within a behavior that is based on cooperation and expectation of resource sharing (Dominguez-Rodrigo & Pickering, 2017; Wood & Gilby, 2019). Primary access to animals (either through hunting or aggressive/confrontational scavenging) by early Homo would have required similar complex behaviors. The evolutionary importance of the adoption of meat in the early Pleistocene hominin diet is therefore twofold: not only did it trigger (or was the outcome of) relevant anatomical and physiological changes, but it also led to significantly different lifestyles and socio-reproductive behaviors, as is reflected in the regular acquisition by early Homo of fully fleshed carcasses. The predominant role of hominins within the predatory guild would ultimately also explain the subsequent demographic and geographic expansion of Homo outside Africa, which is documented pene-contemporaneously with the emergence of the earliest archaeological record in Africa.

Conclusions

A comprehensive taphonomic analysis of the DS archaeofaunal assemblage, including analysis of bone breakage patterns, bone surface modifications, taphotypes, fabric analysis, skeletal representation patterns, and statistical spatial analysis, has strongly supported the hypothesis that early humans at DS had mostly primary access to fleshed carcasses prior to any other carnivore (Cobo-Sánchez, 2020) (see Supplementary Information). This invalidates multi-patterned felid-hominin-hyenid models that assume that hominins played an opportunistic passive role in acquiring carcasses (Blumenschine, 1995). Such models require that felids would have played a major role in the defleshing of carcasses and that they modified bone surfaces distinctively (Domínguez-Rodrigo et al., 2021c). If felids had been the main defleshing agent at DS, their tooth marks should be found on meat-bearing bones and bone portions like long bone midshafts. If non-primate carnivore action has been restricted to post-depositional ravaging by durophagous carnivores, then hyenas would be expected to have impacted those elements and portions. The tooth mark deep learning analysis described in the present work supports the latter option, by showing that the vast majority of identifiable tooth marks are confidently attributed to hyenas and not lions. The reliability of this technique lies in the discovery of only one carcass that bears taphotype traces of felid and hominin modification, and whose tooth marks were identified as felid-made by using similar techniques to those displayed here (Domínguez-Rodrigo et al., 2021c). The fact that felid tooth marks are absent from the rest of carcasses documented at DS, and that typical felid damage (like preferential modification of the medial epicondyle of the distal humerus, the tibial crest, the femoral trochanters, the pelvic iliac crest or the scapular blade and neck) (Parkinson, Plummer & Hartstone-Rose, 2015; Domínguez-Rodrigo, Barba & Egeland, 2007; Domínguez-Rodrigo et al., 2021b) is also not observed in the assemblage reinforces the interpretation of a hominin-hyenid agency in the formation and modification of the DS faunal assemblage. Here, the coincidence in agency attribution by both deep learning models confirms this hypothesis. The results presented here must be confirmed in future research by expanding the experimental sample size, and by adding more carnivore types to the comparison, including extinct taxa. In the meantime, these results add more heuristics to the interpretation of a hominin agency in the accumulation of carcasses and the exploitation of bulk flesh at the site, as inferred from the complete taphonomic analysis of the site (Cobo-Sánchez, 2020).

Supplemental Information

Supplemental Information 1 Summary of the taphonomic analysis of DS and scores for the experimental models

Click here for additional data file.

For their indispensable work in the field research conducted at DS (Olduvai Gorge), we would like to express our appreciation and acknowledgement to: Agness Gidna, Julius Sulley, Lazaro Sarwatt, Yacob Matle, Yona Thomas, Thomas Madangi, Nicolaus Dohho, Caroli Maole, Francis Fabiano, Sangau Letuma, Nicodemus Burra, Ibrahim Mathias, Shabany Bakari, Julia Aramendi, Elia Organista, Ainara Sistiaga, David Martín, and David Uribelarrea. We are also grateful to the Olduvai Gorge Paleoanthropology field school (University of North Carolina Greensboro) and Complutense University students for their involvement in the recovery and cleaning of the fossils. We also sincerely appreciate the work of Aroa Serrano, Laura Gómez Tejerina, Andrea Díaz, Sofía de León and Laura Gómez Morgado for restoring the DS fossil remains. Finally, we are especially thankful to the very useful suggestions made by the anonymous reviewer and Ilkka Sipilä to an earlier draft of this paper.

Additional Information and Declarations

Competing Interests

Author Contributions

Field Study Permissions

Data Availability

The authors declare there are no competing interests

Lucía Cobo-Sánchez conceived and designed the experiments, performed the experiments, analyzed the data, prepared figures and/or tables, authored or reviewed drafts of the article, and approved the final draft.

Marcos Pizarro-Monzo conceived and designed the experiments, performed the experiments, analyzed the data, authored or reviewed drafts of the article, and approved the final draft.

Gabriel Cifuentes-Alcobendas performed the experiments, authored or reviewed drafts of the article, and approved the final draft.

Blanca Jiménez García performed the experiments, authored or reviewed drafts of the article, and approved the final draft.

Natalia Abellán Beltrán performed the experiments, authored or reviewed drafts of the article, and approved the final draft.

Lloyd A. Courtenay conceived and designed the experiments, authored or reviewed drafts of the article, and approved the final draft.

Audax Mabulla conceived and designed the experiments, authored or reviewed drafts of the article, and approved the final draft.

Enrique Baquedano conceived and designed the experiments, authored or reviewed drafts of the article, and approved the final draft.

Manuel Domínguez-Rodrigo conceived and designed the experiments, performed the experiments, analyzed the data, authored or reviewed drafts of the article, and approved the final draft.

The following information was supplied relating to field study approvals (i.e., approving body and any reference numbers):

COSTECH and Tanzanian Department of Antiquities approved the study (2021-631-NA-2006-115).

The following information was supplied regarding data availability:

The data is available at Harvard Dataverse: Cobo Sanchez, Lucia, 2021, ”Replication Data for: Computer vision indicates primary access to meat by early Homo 1.84 million years ago”, https://doi.org/10.7910/DVN/BQTKBA, Harvard Dataverse, V6.

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
