# Peer review of "Computer vision supports primary access to meat by early Homo 1.84 million years ago"

_PeerJ, doi:10.7717/peerj.14148_

## Round 0.1 · original submission · Major Revisions

Dear author. Two reviewers recommended major revisions to your paper. Please address these, and resubmit.

Regards,

Shaw

·

Basic reporting

The article is well written overall and easy to follow, but there are some minor corrections that should be made:

- Lines 49-50: in the very first sentence, a reference to some early publications could be helpful
- Lines 52: ”most commonly limited to flesh scraps and long bone marrow” – do you mean "mostly limited"?
- Lines 65-66: ”This might have been the first recorded impact on humans at an ecosystemic scale” – do you mean ”the first time hominins had an ecosystem-wide impact”?
- Lines 75-76: perhaps consider citing (Domínguez-Rodrigo et al. 2010; McPherron et al. 2010, 2011) as well as (Dominguez-Rodrigo and Alcalá 2019; Domínguez-Rodrigo and Alcalá 2016; Harmand et al. 2015) to provide context.
- Lines 82-83: ”The only evidence provided for this would be the high tooth mark frequencies documented on long bone shafts from the 1.84 Ma site of FLK Zinj (Olduvai Gorge)” – this is quite unclear. Do you mean ”would be the high tooth mark...” or ”is the high tooth mark...”? Is this sentence missing a reference?
- Line 84: the mixing of percentages and ’times’ requires attention
- Line 91: instead of writing ”carnivore-specific identifications could be made”, write ” carnivore-specific identifications were made”
- Line 154: “consisting on” should be “consisting of”
- Lines 181-182: Specify which models are being discussed
- Line 195: what is meant by pairwise comparisons here?
- Lines 201-202: what is meant by the compilation process?

Citations and references to Portalier et al. and Potts are lacking years. The title for Pobiner et al. 2020 is in capital letters, so reformat to match the other references.

Please also follow the journal headings: Introduction, Materials & Methods, Results, Discussion, Conclusions.

Experimental design

The article constitutes an original primary research and fits within the scope of the journal. The research question is well defined, relevant and meaningful. The submitted paper also aims to fill an important gap in our knowledge of human evolution.

In lines 197-198, the authors state that "The last fully connected layer of the network used a “sigmoid” activation for the binary comparison between lions and spotted hyenas". However, the output of sigmoid function is the probability that the input data represents the positive class (i.e. class with a label of 1). Thus, when the authors state on lines 222-224 that "Marks were interpreted as agent specific only when the two models coincided in the determination and, at least one of them, yielded classification probabilities >70%", it seems to me as though the results of the study are pre-determined to select for one of the two classes. I presume this class to be hyaena since most of the predictions indicate hyaena tooth marks. The authors may therefore be confusing the confidence scores produced by softmax function and the probability scores produced by sigmoid function. Also note that softmax function does not produce true probability scores as deep neural networks are miscalibrated (see Guo et al. 2017). It would be easier to evaluate what the authors have done if the code was made available, and it could well be that this is simply an issue of clarity. Finally, it appears as though the authors describe an ensemble modelling method (majority-voting) on these lines (221-225), so it would be good to make this explicit.

Currently, the article does not contain enough information about the models and the training process for the reader to trust these models. For example, there is no evidence to show that the model is not over- or under-fit nor have the authors included evidence to show that their models use the tooth marks to identify the species. Thus, it would be beneficial to the article if the authors included (as a supplementary file) more details about how the validation and training losses and accuracies evolved during the model training. Likewise, the authors should use a saliency method of their choice (as they have done elsewhere, e.g. Jiménez-García et al. 2020) to provide evidence (again, as a supplement) that the models truly classified the tooth marks based on the tooth marks themselves and not some circumstantial information. Moreover, it would be a good idea for the authors to incorporate an early stopping mechanism (e.g. stop training if validation loss has not improved in 5 epochs) for their model training as it would help with preventing over-fitting (see Prechelt 2012).

Furthermore, nowhere is it mentioned how big the image dataset is nor is it stated how many images were included for each class. This is an issue especially when discussing accuracies in a binary classification. Consider for instance a highly imbalanced dataset where class A consists of 990 samples and 10 are in class B. If the model correctly classifies all samples in class A but none in Class B, the accuracy is 99% but the model will be a completely useless model as it can only classify class A items. Thus, while it is good that F1-scores are shown in the tables, please include numeric values for class sizes and dataset in general. Another important aspect of reporting classification results is the inclusion of a confusion matrix, so please add one. Finally, please use micro-average F1-scores if the two classes are highly imbalanced since neither class is more important than the other a priori, but the number of specimens included in each class matters an awful lot and has an impact on the overall model F1-scores. Alternatively, you may find it more informative in general to provide model accuracy, precision, recall, and F1-scores for both species separately.

Validity of the findings

The ideas conveyed in the article are very interesting and make for a thought-provoking read that adds to the growing archaeological literature where deep learning methods are applied. The article is also a good follow up on Abellán et al.'s (2021) ideas. While I commend the authors for using deep learning methods to solve archaeological problems, there is one large question mark that hangs over this paper regarding the validity of the findings: how does the dental morphology of modern lions and hyaenas compare to their counterparts at David’s Site, nearly two million years ago? The issue with using modern animals’ tooth marks to train deep learning CNNs is that the models will only learn to classify tooth marks that fall within the variation of the modern samples – no amount of data augmentation can help with this. Thus, if Early Pleistocene lion and hyaena dental morphologies do not fall within the variation of modern lion and hyaena, the conclusions drawn in this article are not valid even if the models were (which cannot be judged based on the current information). This aspect of the study should be at the very least mentioned in the article as a possible caveat as I can understand that this research may not be available. If it is available, it would be important to provide references showing that past and present lions’ and hyaenas’ dental morphologies fall within the same variance.

Regarding reproducibility, although the data is made available, I would recommend the authors to include links to a Github or some other code repository for others to follow and replicate their image classification methodology in full. This would help the reviewers as well as other readers who wish to perform similar tasks.

The Discussion in the article is detailed and logical, but it hinges on the results of the deep learning methodology, which is not convincing enough in its current state. The article also needs a clear Conclusions section.

Additional comments

Overall, the information provided in the supplementary information regarding skeletal part representation and bone breakage patterns was more convincing than the deep learning methodology regarding the main argument, which is that humans had primary access to meat through hunting. I would suggest bringing this line of argument to the fore and focusing less on the deep learning.

I'm also providing here the full references to the sources I've cited above:
Abellán, N., Jiménez-García, B., Aznarte, J., Baquedano, E., & Domínguez-Rodrigo, M. (2021). Deep learning classification of tooth scores made by different carnivores: achieving high accuracy when comparing African carnivore taxa and testing the hominin shift in the balance of power. Archaeological and Anthropological Sciences, 13(31), 14. https://doi.org/10.1007/s12520-021-01273-9
Dominguez-Rodrigo, M., & Alcalá, L. (2019). Pliocene Archaeology at Lomekwi 3? New Evidence Fuels More Skepticism. Journal of African Archaeology, 17(2), 173–176. https://doi.org/10.1163/21915784-20190006
Domínguez-Rodrigo, M., & Alcalá, L. (2016). 3.3-Million-Year-Old Stone Tools and Butchery Traces? More Evidence Needed. PaleoAnthropology, 46–53. https://doi.org/10.4207/PA.2016.ART99
Domínguez-Rodrigo, M., Pickering, T. R., & Bunn, H. T. (2010). Configurational approach to identifying the earliest hominin butchers. Proceedings of the National Academy of Sciences of the United States of America, 107(49), 20929–20934. https://doi.org/10.1073/pnas.1013711107
Guo, C., Pleiss, G., Sun, Y., & Weinberger, K. Q. (2017). On Calibration of Modern Neural Networks. In D. Precup & Y. W. Teh (Eds.), Proceedings of the 34th International Conference on Machine Learning (Vol. 70, pp. 1321–1330). PMLR.
Harmand, S., Lewis, J. E., Feibel, C. S., Lepre, C. J., Prat, S., Lenoble, A., et al. (2015). 3.3-million-year-old stone tools from Lomekwi 3, West Turkana, Kenya. Nature, 521(7552), 310–315. https://doi.org/10.1038/nature14464
Jiménez-García, B., Aznarte, J., Abellán, N., Baquedano, E., & Domínguez-Rodrigo, M. (2020). Deep learning improves taphonomic resolution: high accuracy in differentiating tooth marks made by lions and jaguars. Journal of the Royal Society Interface, 17(168), 8.
McPherron, S. P., Alemseged, Z., Marean, C. W., Wynn, J. G., Reed, D., Geraads, D., et al. (2010). Evidence for stone-tool-assisted consumption of animal tissues before 3.39 million years ago at Dikika, Ethiopia. Nature, 466(7308), 857–860. https://doi.org/10.1038/nature09248
McPherron, S. P., Alemseged, Z., Marean, C., Wynn, J. G., Reed, D., Geraads, D., et al. (2011). Tool-marked bones from before the Oldowan change the paradigm. Proceedings of the National Academy of Sciences of the United States of America, 108(21), E116. https://doi.org/10.1073/pnas.1101298108
Prechelt, L. (2012). Early stopping - But when? In G. Montavon, G. B. Orr, & K. Müller (Eds.), Neural Networks: Tricks of the Trade. Lecture Notes in Computer Science, vol 7700 (2nd ed., pp. 53–67). Berlin Heidelberg: Springer-Verlag. https://doi.org/10.1007/978-3-642-35289-8_5

Reviewer 2 ·

Basic reporting

Thank you for allowing me to review the manuscript “Computer vision indicates primary access to meat by early Homo 1.84 million years ago” written by Lucía Cobo-Sánchez and collaborators. The paper presents the taphonomic analysis of carnivore tooth marks from the DS site, Olduvai, Tanzania, through the application of deep learning methods. Overall, the paper is of good quality (appropriate methodology, good comparative sample size, good knowledge and background). However, I have two main concerns about the manuscript: (1) The main reasoning and argument of the paper are not appropriate to answer the question of hominin subsistence behaviours; (2) The title of the paper does not reflect the content.
I agree with most of the conclusions and findings and think that every argument needed is there but would greatly benefit from a deep restructuration (see 3. Experimental design). By rearrange the supplementary and original manuscript, the results will better answer the question and will fit the title.
Therefore, I recommend several modifications. Ultimately, I think that this paper could be accepted for publication after considering the comments below.

Experimental design

The research hypotheses are well defined, lines 95-98: “If hominins were preceded by lions in carcass consumption, we would expect the tooth mark signals on meat-bearing long bone shafts to be diagnostically felid. If, instead, hominins had primary access and hyenas intervened postdepositionally, then a hyenid signal on the same bone portions would be expected.”
The main problem I have with the paper is about the original statement and on its interpretation. Indeed, if hominins were preceded by lions in carcass consumption, we would expect tooth marks on meat-bearing bones to be diagnostically felid. But, it is also highly probable that the shallow tooth marks made by felids in the first place have been hidden by high fragmentation of the bone assemblage, which seems to be the case here (Figure S2e and Figure S4d). Moreover, it is not because there is an absence of felid tooth marks that hominins had primary access to the carcasses. From my point of view, it is not possible to use the presence or absence of certain type of tooth marks to infer hominin subsistence behaviours. The material is then not appropriate to answer the original question.
Showing the apparent absence of felid modifications on the bone assemblage is indeed important and part of the taphonomic history and can be one piece of evidence in order to reconstruct hominin subsistence behaviour. But it cannot be used by itself. All the taphonomic information that the reader needs are in the supplementary material. We don’t have access to mandatory raw data (such as the distribution of cutmarks and percussion marks along each bone elements and by size classes). The Figure S5 and S6, for example, are much more accurate than the absence of felid tooth marks in order to argue for an early access to carcasses by hominins.
Thus, I would recommend introducing most of the taphonomic data from the Supplementary material into the actual manuscript, and maybe to put most of the deep learning methods and results into the Supplementary material. In that state, the supplementary material is much more accurate than the actual draft in order to answer the original question and to fit with the title.

Validity of the findings

The findings are of much interest for the taphonomic community. It is a good example of multivariate taphonomic approach (only if we consider the Supplementary material). The Figure S5 and S6 are particularly helpful and accurate. There could be improved by being integrated into the original manuscript and by putting all the raw data used to produce these figures in the Supplementary.

Additional comments

Introduction
Line 57: “(Potts; Blumenschine, 1991)” Incomplete reference (see also the Bibliography section below)
Lines 57-59: “Hunting would also have provided a regular source of high-quality food, which would have prompted the transfer of basal metabolic energy from the digestive tract to the brain enabling its evolution (Aiello & Wheeler, 1995)”. Even if it could be true, this hypothesis still lack evidence and is still a debate today (Navarrete et al., 2011; Isler and van Schaik, 2012; Cornélio et al. 2016; Daujeard & Prat, 2022).
Lines 61-62: “Ecologically, it has also been argued that the genus Homo started to impact African biomes by 2 Ma by outcompeting other carnivores” This statement needs references.
Lines 87-89: “Subsequent re-examination of the Zinj assemblage led some authors to document that a high proportion of such marks were biochemical bioerosive modifications (Domínguez-Rodrigo & Barba, 2006; Parkinson, 2018).” Then, it appears necessary to add bioerosive modifications into the dataset if they are sometimes identified as tooth marks.
Lines 95-98: “If hominins were preceded by lions in carcass consumption, we would expect the tooth mark signals on meat-bearing long bone shafts to be diagnostically felid. If, instead, hominins had primary access and hyenas intervened postdepositionally, then a hyenid signal on the same bone portions would be expected.” These two sentences state the two null hypotheses of the study. However, I have concerns about their accuracy and about the reasoning behind.
“If hominins were preceded by lions in carcass consumption, we would expect the tooth mark signals on meat-bearing long bone shafts to be diagnostically felid”: This is true indeed. However, identifying tooth marks as felid or hyenid do not directly highlight the carcass consumption nor the access time by hominins.
Moreover, it implies that the bone accumulation at DS is the result of either carnivore or hominin and nothing else. Even by identifying felid tooth marks on long bones, it will not support or discard a primary or secondary access to the carcasses by hominins.
Lines 123-124: “Long bones were also systematically demarrowed by hominins as evidenced by the frequency (6.7%) and distribution of percussion marks” We cannot speak about “systematically demarrowed” for an assemblage with only about 7% of percussion marks.
Lines 132-134: “Here, we will study the 43 tooth marks (15 tooth scores and 28 tooth pits) documented on the bone specimens from DS and provide identification of the carnivore agent(s) involved and their impact.” This should be moved into the Material section, line 168. Moreover, it is not specified on how many bone specimens these 43 tooth marks have been observed. Are the 43 tooth marks located on 43 bone elements?

Methods
Line 157: “lion kills under thr research permit” change “thr” to “the”
Line 209: “Images of BSM were produced with a binocular microscope” change to “Images of bone surface modifications (BSM) were produced…”
Line 215: “differed(Abellán et al., 2021).” Add a space after “differed”.

Discussion
Lines 256-259: “The emergence of early Homo has mainly been linked to two phenomena: encephalization and dietary change. The reduction of the dentition in some hominin taxa for the first time in four million years of evolution could be a reflection of the increase of the food quality which in a savanna biome, was probably the outcome of the adoption of carnivory.” This statement needs to be supported by references. Moreover, it is actually fare more complex than that as “it has been shown that brain enlargement and dental reduction were decoupled and evolved at different rates (Gómez-Robles et al., 2017). Moreover, the morphology of the masticatory system, in particular dentition, does not represent the diet, but reflects what individuals were able to eat rather than what they ate. This is Liem’s paradox (Liem, 1980), of which Paranthropus is a good illustration, with its apparent ambiguity between anatomy and diet.” (Daujeard & Prat, 2022:8).
Line 274: “This is the species that we have in mind when we attribute this archeological record to early Homo.” I would recommend removing “early”, as H. ergaster is definitely not an early representant of the genus Homo, and because DS site is dated around 1.84 Ma, which is actually 1 Ma younger than the first Homo fossil record.
Lines 281-283: “The fact that only hyenas are securely identified as the main bone modifiers in addition to hominins at DS supports that their intervention took place after hominins discarded the carcasses that they consumed.” Why? I understand that hyenas are now securely identified, in contrast with felids, but if there is no co-occurrence on the same specimen of both hominins and hyenas BSM, it remains difficult to argue that hominins and hyenas were interacting with each other. There could have been, for example, two independent accumulations by hominins and hyenas, both by primary access, or hyena after lion. This kind of accumulation can probably happen in a relatively short period of time.
Lines 288-289: “Moreover, the documented taxa diversity at the site contrasts with a highly specialized felid predatory range” I disagree for that statement. Indeed, the diversity observed on bovid material from DS could also has been created by felids in a succession of multiple accumulation event. Especially if we consider extinct felid species, that could have been able to create a large diversity of taphonomic patterns. Moreover, the overrepresentation of Kobus sigmoidalis is significant (%MNI = 39%). As about the size of Kobus leche (Gentry, 2010), this extinct species probably occupied the same kind of environment than the waterbuck or lechwe. These two species inhabit humid woodland environments, and are mostly preyed by felids (lion and leopard) as well as wild dogs (Mitchelle et al., 1965, African Zoology). Finally, there is no faunal list available for DS, expect the one in the Supplementary Information (Figure S3). However, the authors wrote “The bovid dental remains found are DS 22B….. (Figure S3)”. It means that we have no information on the rest of the fauna (especially the carnivore fauna).
Lines 296-297: “This is of extreme relevance because it also shows that hominins occasionally engaged in opportunistic carcass exploitation, like modern human foragers and mammal predators do” Unless if there is definitive hominin damage observed on this specimen (DS4288), there is no reason to think that it has been exploited by hominins after felids, but only by felids. It is not possible to infer hominin subsistence behaviour by using specimens only modified by carnivores.
Lines 305-309: “The only tooth mark from the DS sample used here that was classified as lion-made was interestingly found on a mandible fragment in spatial association to the scavenged radius-ulna. This is important, because it shows that out of the 560 m2 of the excavated site, all lion-impacted bone specimens are in close spatial association.” The authors have to clarified what they mean by “close spatial association”. As there are very few figures in the manuscript, I would recommend the authors to add a schematic plan of the site showing the distribution of the 43 tooth-marked specimens. Because part of the discussion and argument is based on this association, it is mandatory to support this argument with proper archaeological evidence.
Lines 310-312: “Additionally, this also documents that the site underwent virtually very limited post-depositional disturbance, since the original spatial association of the scavenged hominin-processed carcass remains was not altered after discard.” It is an overstatement to argue for an undisturbed bone assemblage based on the assumption that two bones are in “spatial association” and both display felid tooth marks. Even if they belong to the same individual, having two associated specimens among more than 15,000 bone remains is not sufficient to claim for an undisturbed assemblage. I do not say that the assemblage is disturbed, but that this statement will need more evidence to be supported.
Lines 314-319: “The overwhelming confirmation that most tooth marks at the site were made by hyenas and the lack of felid taphotypes on the preserved long bone ends show that hominins were predominantly enjoying primary access to completely fleshed carcasses. The abundance and anatomical distribution of cut marks at DS, which coincide with experimental models of primary and early access to carcasses by hominins, further supports this interpretation.” These sentences need to be rewrite. Indeed, the absence of felid taphotypes and the identification of most tooth marks as hyenid do not indicate primary access by hominins. It just indicates that felids were just a minor taphonomic agent at DS. However, the distribution of cut marks does support an early access to the carcass. In other words, you first discard felid as major taphonomic agent and then demonstrate an early access to the carcass based on direct taphonomic evidence (cut marks). But the tooth marks themselves do not indicate anything about hominin subsistence behaviours. As it is stated in Domínguez-Rodrigo et al. (2007:22):
“Domínguez-Rodrigo et al. (in press) demonstrate that if carnivores and hominids are accumulating and modifying carcasses independently of each other (i.e., they are not transporting or consuming the same carcasses), tooth mark frequencies may mimic “carnivore-first” scenarios, despite the fact that this says nothing about hominid access to carcasses.”
It means that, by identifying tooth marks as hyenid origin, and by eliminating felids as one of the main taphonomic agent, the authors say nothing about hominid access to carcasses. It is only through hominin BSM that it could be achieved. Unfortunately, all of this information is in the supplementary material.
Line 324-325: “Meat must have therefore been crucial for the adaptation of early Homo and must have had serious repercussions for early human physiology.” Why? There is no transition with the previous paragraph. Moreover, the study does not show that “meat must have therefore been crucial for the adaptation of early Homo”. See the previous comment on Line 274 about early Homo.
Lines 328-332: “As a matter of fact, it has been argued that evidence of some typically-human pathologies (like porotic hyperostosis) documented in an infantile 1.5 Ma individual might be related to deficiencies in cobalamin consumption, which could be related to a meat-dependent physiology at that time (Domínguez-Rodrigo et al., 2012)” I also suggest to add “D’Anastasio R, Zipfel B, Moggi-Cecchi J, Stanyon R, Capasso L (2009) Possible Brucellosis in an Early Hominin Skeleton from Sterkfontein, South Africa. PLoS ONE 4(7): e6439. doi:10.1371/journal.pone.0006439” as it is one of the rare direct evidence of occasional meat-eating by Australopithecus.
Lines 332-335: “This first direct evidence of a hominin-hyenid interaction in the modification of an early Pleistocene assemblage also shows that the significant increase in brain size documented after 2 Ma co-occurs with this evidence of bulk flesh consumption by hominins.” I have several issues with that sentence. 1) There is no link between this sentence and the previous one. 2) Does the study show a hominin-hyenid interaction? I don’t think so, unless you have bone specimens showing both hominin and hyenid modifications. Anyway, the authors will have to describe these specimens in order to reconstruct the taphonomic history, but this is not the topic of the paper. So, the deep learning method just helped to identify taphonomic agent, but definitely does not provide any clue on the interaction between hyenid and hominin. 3) As stated in a previous comment the correlation between brain size, chronology and diet is still unclear (Navarrete et al., 2011; Isler and van Schaik, 2012; Cornélio et al. 2016; Daujeard & Prat, 2022).

Bibliography
Lines 404-405: “Antón SC, Potts R, Aiello LC. 2014. Evolution of early Homo: An integrated biological perspective. Science.” Put Homo in italic. Incomplete reference (issue and pages).
Lines 413-415: “Baquedano, E., Domínguez-Rodrigo, M., Musiba, C., 2012. An experimental study of large mammal bone modification by crocodiles and its bearing on the interpretation of crocodile predation at FLK Zinj and FLK NN3. Journal of Archaeological Science 39, 1728-1737.” Put the name journal in italic.
Lines 421-423: “Blumenschine RJ. 1995. Percussion marks, tooth marks, and experimental determinations of the timing of hominid and carnivore access to long bones at FLK Zinjanthropus, Olduvai Gorge, Tanzania. Journal of human evolution 29:21-51.” Put Zinjanthropus in italic.
Lines 424-426: “Capaldo SD. 1997. Experimental determinations of carcass processing by Plio-Pleistocene hominids and carnivores at FLK 22 (Zinjanthropus). Olduvai Gorge, Tanzania. Journal of human evolution 33:555-597.” Put Zinjanthropus in italic.
Lines 512-515: “Hammond AS, Mavuso SS, Biernat M, Braun DR, Jinnah Z, Kuo S, Melaku S, Wemanya SN, Ndiema EK, Patterson DB, Uno KT, Palcu DV. 2021. New hominin remains and revised context from the earliest Homo erectus locality in East Turkana, Kenya. Nature communications 12:1939.” Put Homo erectus in italic.
Lines 518-523: “Herries AIR, Martin JM, Leece AB, Adams JW, Boschian G, Joannes-Boyau R, Edwards TR, Mallett T, Massey J, Murszewski A, Neubauer S, Pickering R, Strait DS, Armstrong BJ, Baker S, Caruana MV, Denham T, Hellstrom J, Moggi-Cecchi J, Mokobane S, Penzo-Kajewski P, Rovinsky DS, Schwartz GT, Stammers RC, Wilson C, Woodhead J, Menter C. 2020. Contemporaneity of Australopithecus, Paranthropus, and early Homo erectus in South Africa. Science 368. DOI: 10.1126/science.aaw7293.” Put Australopithecus, Paranthropus and Homo erectus in italic.
Lines 537-539: “Organista E, Domínguez-Rodrigo M, Egeland CP, Uribelarrea D, Mabulla A, Baquedano E. 2015. Did Homo erectus kill a Pelorovis herd at BK (Olduvai Gorge)? A taphonomic study of BK5. Archaeological and anthropological sciences:1-24.” Put Homo erectus and Pelorovis in italic.
Lines 555-557: “Pobiner B, Dumouchel L, Parkinson J. 2020. A NEW SEMI-QUANTITATIVE METHOD FOR CODING CARNIVORE CHEWING DAMAGE WITH AN APPLICATION TO MODERN AFRICAN LION-DAMAGED BONES. Palaios 35:302-315.” Put in small letters.
Lines 558-559: “Portalier SMJ, Fussmann GF, Loreau M, Cherif M. The mechanics of predator-prey interactions: first principles of physics predict predator-prey size ratios.” The year of publication is lacking.
Line 560: “Potts R. Early Hominid Activities at Olduvai. AldineTransaction.” Incomplete reference.
Lines 567-569: “Spoor F, Leakey MG, Gathogo PN, Brown FH, Antón SC, McDougall I, Kiarie C, Manthi FK, Leakey LN. 2007. Implications of new early Homo fossils from Ileret, east of Lake Turkana, Kenya. Nature 448:688–691.” Put Homo in italic.
Lines 573-574: “Ungar PS. 2012. Dental Evidence for the Reconstruction of Diet in African Early Homo. Current anthropology 53:S318-S329.” Put Homo in italic.
Lines 577-579: “Villmoare B, Kimbel WH, Seyoum C, Campisano CJ, DiMaggio EN, Rowan J, Braun DR, Arrowsmith JR, Reed KE. 2015. Paleoanthropology. Early Homo at 2.8 Ma from Ledi-Geraru, Afar, Ethiopia. Science 347:1352-1355.” Put Homo in italic.
Lines 582-583: “Wood B, Gilby I. 2019. From Pan to man the hunter: hunting and meat sharing by chimpanzees, humans, and our common ancestor.” Incomplete reference.

Supplementary information
Figure S2. Autochthony and fluvial impact. (e) I understand that the data are displayed by %NISP? We also need the scale below. Are the measurements in millimeters?
Figure S3. Taxa and skeletal part representation: The authors do not specify what is shown in the skeletal part profiles (c). Is it percentage of MNE? Percentage of NISP?

Annotated reviews are not available for download in order to protect the identity of reviewers who chose to remain anonymous.

---

## Round 0.2 · Minor Revisions

Thank you for addressing the comments of the reviewers, and resubmitting. I asked two of the reviewers to look at the paper again, and they made valuable suggestions. I ask that you please address these in the paper, and indicate in the rebuttal letter how you dealt with these comments. Almost there!

·

Basic reporting

First of all, I thank the authors for taking my previous suggestions on board - the text is much clearer and it far easier to follow the line of argument. In general, your literature is extensive and arguments are well backed up. Thank you also for pointing out that code and images were (mostly) made available - I say mostly, as I still could not find all of the archaeological test images of the tooth marks from David's site. Please include a link to them.

- Line 93: The use of percentage and the word ’times’ is still confusing. Write either ”300-500% higher” or ”3-5 times higher”
- Lines 140, 180, 184, 185: As you pointed out in your rebuttal that these numbers are the actual photographs, place them in a separate paragraph – currently you are not explicit enough about the actual deep learning data. My suggestion: ”The modern lion and hyaena tooth mark image dataset used to train our models includes a total of 355 tooth marks. Of these, 195 are scores made by lions and 80 scores made by hyaena. In addition, there are 35 pit marks caused by lions and 45 pits by hyaena. We subsequently test our models on the images of 43 tooth marks (15 scores and 28 pits) recorded on bones from DS.”
- Line 198: Please add the version of the ResNet-50 model – in your code you use v1. Also, please cite the original publications for these models.
- Line 215: Add a paragraph break before ”We are aware that modern lions...”
- 216-218: I realise this may be asking a bit much as it is an on-going study, but is there any more information you can provide? Name dropping the author of this study and saying that it is personal communication could also help (don’t need to add in the reference list).
- Line 219: Instead of ”we will make the assumption”, just write ”we assume”
- Line 221: Again instead of ” If such variation existed, we will make a second assumption” write ”Secondly, if such variation existed, we assume that extinct hyenas would have had tooth morphologies (which ultimately determine tooth mark morphologies) more similar to modern hyenas than to modern lions”
- Line 235: Reword ’The lion-hyena tooth mark comparisons’ to ‘The tooth mark classifications’
- Lines 261-265: What are the two models that are being referred to?
- Line 264-265: The wording seems wrong. Why is anything ’discarded for [from?] the interpretation’?
- Line 265: There is no comparison in majority voting. Simply say that you used majority voting. However, see my recommendations on the experimental design.
- Line 270: Remove extra space between ’to’ and ’binary’
- Line 275: Missing s from ’loss’
- Line 276-277: Write ’over- and underfitting’ instead of ’over-underfitting’
- Lines 289-291: This statement is wrong. It is the accuracy that you should care about more than the loss.
- Line 464: Add a space between ’site,’ and ’as’.
- Lines 483-484: Please rephrase ” Finally,we are especially thankful to the very useful suggestions made by Ikka Sipila to an earlier draft of this paper.” as ”Finally, we are especially thankful to the very useful suggestions made by the anonymous reviewer and Ilkka Sipilä to an earlier draft of this paper.”

In addition:
The captions of Figures 1-4 need spaces between ’F1-scores,’ and ’micro-’. Likewise, remove the redundant space between ’macro-’ and ’average’.

The tables showing the metrics for the models that are placed in the supplementary information should be brought in the actual article.

Experimental design

Beyond the below comments, I found the research question well defined and methods acceptable.

- The ensemble methodology is unclear. In the code (for pits), Random Forest is used, which is completely unnecessary for majority-voting unless there is a specific intention that wasn’t specified in the text.
- Having given further consideration to the methodology, I do not agree on the methodology of assigning agency to the tooth marks, especially regarding its impact on Table 3. Using arbitrary thresholds to discard certain marks gives a skewed view of what is happening. How can the reader be sure that you did not select this threshold after seeing the results in order to fit them to your argument? Had you selected 95% as your threshold instead of 70%, none of the score marks would have been classified by your models. The much clearer methodology of achieving what you want is to take the average of the probabilities given by the models and choose the class with the higher probability. The results will stay the same, you are not discarding anything, and the text is much easier to follow. By doing so, you are also shielding your research from unnecessary criticism. For instance, let’s say VGG-19 gives the following estimates: 0.95 (hyaena) and 0.05 (lion); and ResNet-50 gives: 0.75 (hyaena) and 0.25 (lion). You would then compute the mean probability as: Hyaena = (0.95+0.75)/2 and Lion = (0.05 + 0.25)/2, which gives a probability average of 0.85 for hyaena and 0.15 for lion.
- It must be made explicit in the Methods that you created separate models for the pits and the scores.

Validity of the findings

This is the important bit. Currently, I do not trust your results. However, I believe these issues can be addressed relatively easily as you have shown in previous studies that you are able to apply the methods I suggest.

- Based on your Figures 1 and 2, I would say that the models in Figure 1 pass my scrutiny, but models in Figure 2 do not. I get a very strong feeling of over-fitting especially for the right hand side model in Figure 2, while the left hand model seems to have hit its peak within 20 epochs. Training for another 80 epochs seems unjustified and I suspect your model training would have stopped around 15-20 epochs if you used early stopping. I strongly suggest you go back to your models and implement early stopping. This point about training times applies to models in Figure 1 as well.
- In Figures 3 and 4, over-fitting is clearly present and this is evident in the accuracies which are more important than the loss when evaluating over-fitting. The only model where overfitting is not an issue is the right hand model in Figure 3. Furthermore, there is an increasing separation in the loss values around 40-50 epochs for the left hand models in both Figures 3 and 4.
- Based on this above assessment, it seems that your most trustworthy model is the ResNet-50 with dropout. You need to make sure that the Table 3 results reflect Dropout models, not the over-fit models.
- I cannot trust the results without trusting the models. Based on the accuracy and loss graphs, over-fitting is obvious for VGG-19 and the non-dropout ResNet-50, which invalidates most of your results. Furthermore, I need to see what the neural network is focusing on – the authors’ rebuttal that saliency was done in a previous article is not valid as they have implemented new models (at least dropout models). To be more specific, I want to know for each test image from DS sample what each of the models are looking at. As you have 2 models (VGG-19 and ResNet-50) and 43 test images, I want to see 86 saliency maps. This information can be fit into a one page figure (the images need not be huge, just clear enough to enable evaluation). Moreover, each article needs to stand on its own legs (as is the guidance from PeerJ) and I find it unhelpful to point me to another article for key information such as saliency mapping.
- This said, I do believe that by implementing early stopping and using saliency after retraining the models, this article can make a very strong argument in favour of early hominin hunting.

Additional comments

I thank the authors for taking the time to address the issues of clarity and for vastly improving their argument. I look forward to reading your next revision.

Reviewer 2 ·

Basic reporting

no comment

Experimental design

no comment

Validity of the findings

no comment

Additional comments

I thank the authors for their replies and corrections. They did a good amount of work on this revision and this should be acknowledged. However, I still have a few comments to make.

Discussion
Lines 256-259: “The emergence of early Homo has mainly been linked to two phenomena: encephalization and dietary change. The reduction of the dentition in some hominin taxa for the first time in four million years of evolution could be a reflection of the increase of the food quality which in a savanna biome, was probably the outcome of the adoption of carnivory.” This statement needs to be supported by references. Moreover, it is actually fare more complex than that as “it has been shown that brain enlargement and dental reduction were decoupled and evolved at different rates (Gómez-Robles et al., 2017). Moreover, the morphology of the masticatory system, in particular dentition, does not represent the diet, but reflects what individuals were able to eat rather than what they ate. This is Liem’s paradox (Liem, 1980), of which Paranthropus is a good illustration, with its apparent ambiguity between anatomy and diet.” (Daujeard & Prat, 2022:8).

Reply of the authors: we disagree. There is extensive literature showing the close link between shape and function. Paranthropus is not a good illustration thereof. It actually proves it. Likewise, brain enlargement and dental reduction are evolutionarily not decoupled, since both appear geologically with the earliest members of the genus Homo.

Reply: Indeed, there is extensive literature showing link between shape and function. Paranthropus is actually a good example, as P. boisei and P. robustus share a very similar dental anatomy but still, differ significantly in their diet. Here, the authors state that brain enlargement and dental reduction were not decoupled, only based on their appearance in the fossil record, which is likely to be a result of palaeontological biases. However, by looking at the relation between the two variables in different hominins species, it has been showed that the two phenomena are likely to be decoupled (Gomez-Roblet et al., 2017).


Lines 281-283: “The fact that only hyenas are securely identified as the main bone modifiers in addition to hominins at DS supports that their intervention took place after hominins discarded the carcasses that they consumed.” Why? I understand that hyenas are now securely identified, in contrast with felids, but if there is no co-occurrence on the same specimen of both hominins and hyenas BSM, it remains difficult to argue that hominins and hyenas were interacting with each other. There could have been, for example, two independent accumulations by hominins and hyenas, both by primary access, or hyena after lion. This kind of accumulation can probably happen in a relatively short period of time.

Reply: By reading this commentary I understand this reviewer is not a taphonomist and is not knowledgeable of the extensive experimental literature about how interaction of multiple carnivores and humans can result in specific taphonomic traces. First, the bone portion distribution of percussion marks and tooth marks only fits experimental scenarios where human hammerstone break bones followed by hyena ravaging on long bone ends, producing more midshaft specimens in the process (e.g., Marean, 1991, Marean and Kim 1993; Marean and Bertino, 1994, Capaldo 1995; Arriaza et al., 2019). Secondly, if an independent accumulation between hominins and hyenas existed, this could be easily detected through the distribution of long bone profiles, with the hyena part being mostly cylinders and/or partially broken elements, which could not be mistaken with the minute fragmentation produced by hammerstone breakage. In addition, tooth mark frequencies would be much higher (the frequency of conspicuous and inconspicuous tooth marks in hyena-made accumulations ranges between 80%-100%. If this were not enough, hyenas produce specific hyena taphotypes on long bone ends. None of this is observed at DS.

Reply: It is interesting to see that the authors needed first to demise the reviewer by attacking its skills and knowledge. Besides this argumentum ad personam, you cannot expect the reader to already know everything about the site you are describing here. There is no indication of the absence of hyena taphotypes (such as bone cylinders or typical long bones reduction) in the main text or the supplementary material, although it is stated in the supplementary material that “Overall, the bone breakage assessments applied to DS do not clearly point to a single bone breakage agent and suggest that both hominins and hyenas contributed to fracturing at DS”. So, based on that, it is expect from a reviewer point of view to ask about an alternative scenario where hyenas could have been an independent taphonomic agent on the site, and to ask the authors to clarify this point.


Lines 314-319: “The overwhelming confirmation that most tooth marks at the site were made by hyenas and the lack of felid taphotypes on the preserved long bone ends show that hominins were predominantly enjoying primary access to completely fleshed carcasses. The abundance and anatomical distribution of cut marks at DS, which coincide with experimental models of primary and early access to carcasses by hominins, further supports this interpretation.” These sentences need to be rewrite. Indeed, the absence of felid taphotypes and the identification of most tooth marks as hyenid do not indicate primary access by hominins. It just indicates that felids were just a minor taphonomic agent at DS.

Reply: Wrong. The presence or absence of felid taphotypes has nothing to do with their quantity. They simply show if felids impacted the assemblage regardless of their intensity. Their absence proves they did not. How can the referee justify that felids were a minor agent at DS? Please provide empirical argumentation instead of unsustainable speculation.

Reply: Well, based on a recent publication by Domínguez-Rodrigo et al (2022, Ann. N. Y. Acad. Sci.), it has been shown that the felids were indeed present at the DS site, and that Hominins were probably scavenging after them (at least for that specific bovid carcass). It is therefore logical to consider felids to have been (at least at some points) a minor taphonomic agent at DS site.


Lines 328-332: “As a matter of fact, it has been argued that evidence of some typically-human pathologies (like porotic hyperostosis) documented in an infantile 1.5 Ma individual might be related to deficiencies in cobalamin consumption, which could be related to a meat-dependent physiology at that time (Domínguez-Rodrigo et al., 2012)” I also suggest to add “D’Anastasio R, Zipfel B, Moggi-Cecchi J, Stanyon R, Capasso L (2009) Possible Brucellosis in an Early Hominin Skeleton from Sterkfontein, South Africa. PLoS ONE 4(7): e6439. doi:10.1371/journal.pone.0006439” as it is one of the rare direct evidence of occasional meat-eating by Australopithecus.

Reply: The pathological evidence of brucellosis on the australopithecine specimen is not well justified. We do not believe in it and prefer accordingly not to mention it.

Reply: Well, because the authors used palaeopathologies to discuss meat-eating behaviours among hominins here, I think that it is appropriate to ask for the proper citations and indicate the reader that there is not only one reference about that.

---

## Round 0.3 · accepted · Accept

Thank you for addressing all the comments of the reviewers.